# Cross continental increase in methane ebullition under climate change

Ralf C.H. Aben [1,2], Nathan Barros [3], Ellen van Donk[2,4], Thijs Frenken [2], Sabine Hilt[5], Garabet Kazanjian[5], Leon P.M. Lamers[1,6], Edwin T.H.M. Peeters [7], Jan G.M. Roelofs[1,6], Lisette N. de Senerpont Domis[2,7], Susanne Stephan [8], Mandy Velthuis [2], Dedmer B. Van de Waal [2], Martin Wik[9], Brett F. Thornton[9], Jeremy Wilkinson[10], Tonya DelSontro[11] & Sarian Kosten [1,2]

Methane ($CH_4$) strongly contributes to observed global warming. As natural $CH_4$ emissions mainly originate from wet ecosystems, it is important to unravel how climate change may affect these emissions. This is especially true for ebullition (bubble flux from sediments), a pathway that has long been underestimated but generally dominates emissions. Here we show a remarkably strong relationship between $CH_4$ ebullition and temperature across a wide range of freshwater ecosystems on different continents using multi-seasonal $CH_4$ ebullition data from the literature. As these temperature–ebullition relationships may have been affected by seasonal variation in organic matter availability, we also conducted a controlled year-round mesocosm experiment. Here 4 °C warming led to 51% higher total annual $CH_4$ ebullition, while diffusion was not affected. Our combined findings suggest that global warming will strongly enhance freshwater $CH_4$ emissions through a disproportional increase in ebullition (6–20% per 1 °C increase), contributing to global warming.

[1] Department of Aquatic Ecology and Environmental Biology, Institute for Water and Wetland Research, Radboud University, P.O. Box 9010, 6500 GL Nijmegen, The Netherlands. [2] Department of Aquatic Ecology, Netherlands Institute of Ecology (NIOO-KNAW), P.O. Box 50, 6708 PB Wageningen, The Netherlands. [3] Federal University of Juiz de Fora, Juiz de Fora, Minas Gerais 36036-900, Brazil. [4] Department of Ecology and Biodiversity, University of Utrecht, P.O. Box 80.056, 3508 TB Utrecht, The Netherlands. [5] Department of Ecosystem Research, Leibniz-Institute of Freshwater Ecology and Inland Fisheries, Müggelseedamm 301, 12587 Berlin, Germany. [6] B-WARE Research Centre, P.O. Box 6558, 6503 GB Nijmegen, The Netherlands. [7] Department of Aquatic Ecology and Water Quality Management, Wageningen University, P.O. Box 47, 6708 PB Wageningen, The Netherlands. [8] Department of Experimental Limnology, Leibniz-Institute of Freshwater Ecology and Inland Fisheries, Alte Fischerhütte 2, OT Neuglobsow, 16775 Stechlin, Germany. [9] Department of Geological Sciences and Bolin Centre for Climate Research, Stockholm University, Stockholm SE-10691, Sweden. [10] University of Koblenz-Landau, Institute for Environmental Sciences, Fortstr. 7, 76829 Landau, Germany. [11] Groupe de Recherche Interuniversitaire en Limnologie (GRIL), Département des Sciences Biologiques, Université du Québec à Montréal, Montréal, Canada H3C 3P8 QC. Correspondence and requests for materials should be addressed to S.K. (email: S.Kosten@science.ru.nl)

Despite their small global extent (only ca. 4% of land area)[1,2], freshwater ecosystems are important drivers of the global greenhouse gas (GHG) balance[3]. Global freshwater $CH_4$ emissions have been estimated to correspond to at least 25% (in $CO_2$ equivalents) of the terrestrial GHG sink[4]. Broad-scale emission estimates are, however, biased because most studies focus on diffusive fluxes and neglect the large emission component of ebullition[4,5]. Because ebullition often forms the dominant emission pathway, this may lead to a large underestimate of freshwater $CH_4$ emissions[4,5]. Accurate estimates are still hampered by a scarcity of reliable ebullition data caused largely by the extremely heterogeneous occurrence of ebullition within a system, both in space and in time[4,6–9].

Anaerobic mineralization of sediment organic matter, the source of ebullitive $CH_4$ fluxes, tends to increase exponentially with temperature, forming a potential positive feedback to global warming[10]. Net $CH_4$ emissions from freshwaters to the atmosphere, however, are usually the result of microbial $CH_4$ production and consumption. $CH_4$ consumption mainly takes place in the oxic sediment top layer and water column, and can consume up to 100% of dissolved $CH_4$[11]. Like production, consumption rates also increase with temperature[12–14]. Nonetheless, if a warming-induced increase in $CH_4$ production is concomitant with a constant or higher fraction of $CH_4$ escaping consumption, e.g., through plants or ebullition, an increase in $CH_4$ emission due to global warming is to be expected. To date, a positive relationship between temperature and ebullition has been observed in a number of high-latitude systems[15,16], but there is currently no general consensus that this is a global phenomenon. Also, it is not known whether this response is due to direct effects of temperature (e.g., on microbial process rates) or collinear indirect effects (e.g., enhanced substrate availability due to increased primary production and sedimentation of organic substrates).

To quantify the increase in $CH_4$ ebullition due to temperature-induced increase in sediment $CH_4$ production, we combined a standardized search for published data (as in ref. [17]) with a mesocosm experiment. In the literature search, we only included studies with ebullition data from different types of freshwater ecosystems that covered a temperature range of at least 10 °C, and excluded short-term (<24 h), infrequent (<once per month) ebullition measurements since these likely underestimate the

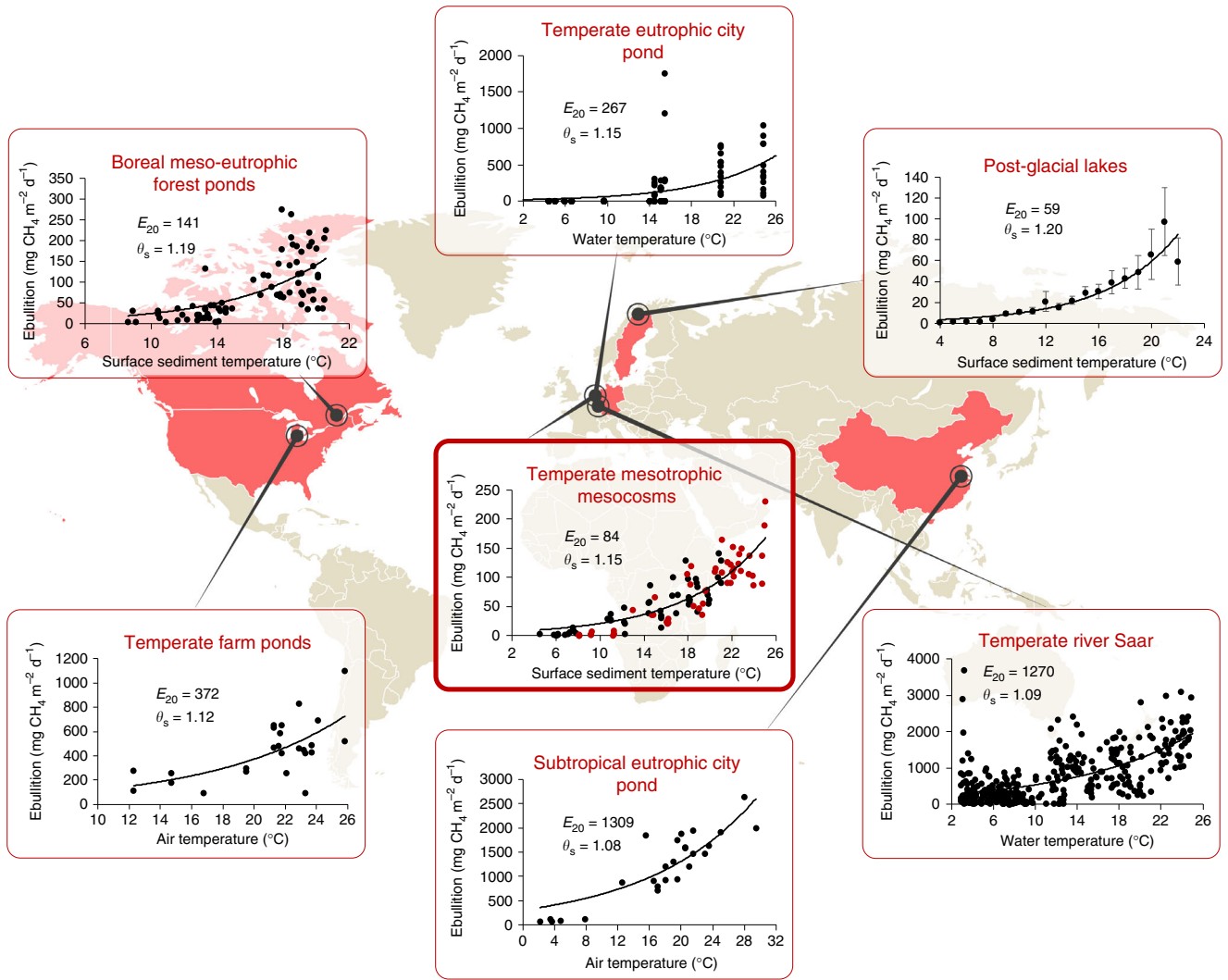

**Fig. 1** Relationships between temperature and $CH_4$ ebullition found in different types of freshwater ecosystems. Regression lines represent the fitted modified Arrhenius expression; see Eq. (2). Error bars denote 95% confidence intervals (Post-glacial lakes). Graphs with thin border lines represent field data. Graph with thick border line represents experimental data. For the latter, black and red dots represent control and warm treatment, respectively. For studies with multiple data sets, the location with the longest data record is depicted. See Table 1 and Supplementary Table 1 for info on all data sets. Details on the origin and acquisition of data are described in the "Methods" section

**Table 1 Temperature dependence of CH₄ ebullition in different freshwater ecosystems**

| System | $E_{20}$ (95% CI) | Overall $\theta_s$ (95% CI) | No. of observations ($n$) |
|---|---|---|---|
| Subtropical eutrophic city pond (D1) | 744 (620–868) | 1.08 (1.05–1.11) | 26 |
| Subtropical eutrophic city pond (D2) | 953 (736–1171) | 1.12 (1.03–1.21) | 16 |
| Subtropical eutrophic city pond (D3)[a] | 1309 (1146–1472) | 1.08 (1.05–1.10) | 26 |
| Post-glacial lakes[a] | 59 (54–65) | 1.20 (1.18–1.22) | 10,227 |
| Temperate river Saar (ABT1) | 3015 (2828–3202) | 1.07 (1.06–1.08) | 291 |
| Temperate river Saar (ABT2) | 1158 (997–1319) | 1.06 (1.04–1.08) | 311 |
| Temperate river Saar (ABT3) | 1813 (1692–1933) | 1.07 (1.06–1.08) | 259 |
| Temperate river Saar (ABT4)[a] | 1270 (1186–1354) | 1.09 (1.08–1.10) | 342 |
| Boreal meso-eutrophic forest ponds[a] | 141 (119–163) | 1.19 (1.11–1.27) | 77 |
| Temperate eutrophic city pond[a] | 267 (209–325) | 1.15 (1.10–1.21) | 132 |
| Temperate farm ponds[a] | 372 (274–470) | 1.12 (1.04–1.20) | 25 |
| Mesocosm experiment[a] | 84 (78–90) | 1.15 (1.13–1.17) | 104 |

$E_{20}$ represents the modelled CH₄ ebullition at 20 °C (mg m$^{-2}$ d$^{-1}$) and $\theta_s$ the overall system temperature coefficient; see Eq. (2). Regressions were significant in all analyses ($P < 0.001$). For characteristics of the systems and corresponding references, see Supplementary Table 1. Details on the origin and acquisition of data are described in the "Methods" section
[a]Systems presented in Fig. 1

ebullitive flux[7,9]. To exclude influences of confounding factors occurring in the field, and to further unravel the relative importance of the different processes responsible for the temperature-induced increase in ebullition, we conducted an experiment in 1000 L mesocosms containing natural lake sediments and plankton communities comparing a temperate (control; $n = 4$) and a warming (+4 °C; $n = 4$) scenario (IPCC scenario RCP8.5[18]). Since the frequency of heat waves is expected to increase over most land areas under future climate projections[18], we included a midsummer 7-day heat wave (+4 °C). Day and night water-atmosphere diffusive gas fluxes were measured once every 2 weeks, using a closed chamber connected to a GHG analyzer, whereas bubbles were collected continuously by inverted funnel-type bubble traps. Our data compilation shows remarkably strong relationships between CH₄ ebullition and temperature across a wide range of freshwater ecosystems, which we also observed in our mesocosm experiment. Our analysis of the experimental data further suggests that direct effects of temperature on microbial growth and metabolism are driving the increase in ebullition. These combined findings indicate a positive feedback for global warming.

## Results

**Temperature drives ebullition on a large spatial scale.** Analysis of the field data meeting our criteria reveals strong positive relationships between CH₄ ebullition and temperature for a wide range of shallow, freshwater ecosystems in Asia, North America, and Europe (Fig. 1 and Table 1). We observed large differences in the magnitude of ebullition among the different freshwater ecosystems (Fig. 1). Although methodological differences in determining ebullition rates play a role here, differences in quantity and quality of sediment organic matter[19–24], sediment structure[25], and differences in the availability of nutrients, oxygen, and alternative electron acceptors[19] are also known to affect the magnitude of ebullition. Freshwaters showing high primary production and those that receive substantial loads of allochthonous carbon are more likely to have high CH₄ ebullition rates[15,23,25–27], while systems with low primary production tend to have low ebullition rates and also may have an obscured temperature effect on ebullition rates, likely due to substrate limitation[15].

The temperature–ebullition relationships from field measurements are likely to be influenced by differences in sampling methods as well. The available temperature data, for instance, was measured in air, water, or sediment. Also the period of sampling varied (year-round versus spring–summer or summer–autumn),

which may have influenced the temperature coefficient in multiple ways. For one, a decrease in CH₄ solubility in the pore water with rising temperatures will lead to a stronger increase in ebullition during the spring–summer period, whereas the opposite may occur when temperatures drop (see below for a quantitative analysis). Second, seasonal variations in organic carbon supply and oxygen availability alter CH₄ production[19,21,28,29]. Finally, fluctuations in forcing mechanisms such as shear stress, atmospheric pressure, and hydrostatic pressure[7,30] (such as the passage of ships in the River Saar[7]) may have weakened the temperature–ebullition relationship. Still, despite these putative confounding factors, ebullition was strongly related to temperature in the systems we analyzed (Table 1).

**Drivers of increased ebullition under warming.** CH₄ ebullition showed a strong relationship with temperature in our mesocosm experiment (Supplementary Fig. 1), which fits our analyses of data from field measurements (Fig. 1 and Table 1). We found that cumulative annual ebullition was 51% higher in the warm treatment ($P = 0.009$; Fig. 2), substantiating that enhanced CH₄ ebullition in temperate regions is a realistic future scenario. Warming may affect CH₄ ebullition in different ways. System productivity, for example, may be affected by warming[31–33], altering the availability of substrate for methanogenesis. We found, however, no significant differences in gross primary production nor in sedimented carbon between our control and warm treatment (Supplementary Fig. 2), indicating that substrate supply was not responsible for the large increase in CH₄ ebullition with warming. We estimate that the effect of temperature-dependent changes in CH₄ solubility during the spring–midsummer period (period with increasing temperature) contributed 14% (control) and 7% (warm) to the total CH₄ ebullition as a result of dissolved CH₄ that turned gaseous. For the midsummer–winter period (period with decreasing temperature), CH₄ ebullition was lowered by 17% (control) and 13% (warm) as a result of increased pore water solubility (Supplementary Fig. 3). However, effects of changes in CH₄ solubility were negligible for cumulative annual ebullitive fluxes. The negligible difference in organic carbon supply as well as the negligible effect of changes in CH₄ solubility strongly suggests that temperature dependency of CH₄ ebullition in our experiment was mostly driven by enhanced microbial metabolism and growth.

**Effects of increased temperature on diffusive CH₄ emissions.** Besides emission by ebullition, CH₄ may also diffuse from the water column into the atmosphere. Diffusive CH₄ emissions in

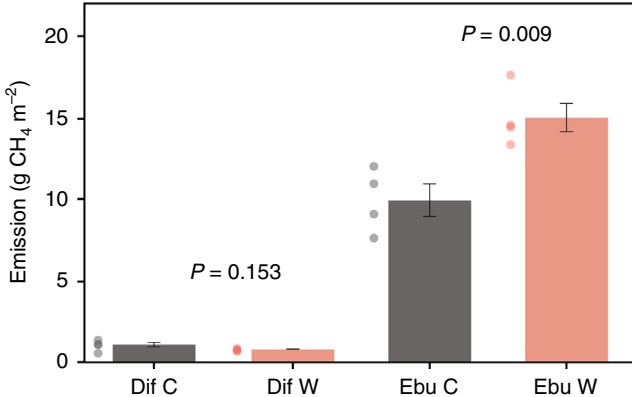

**Fig. 2** Cumulative annual diffusive (Dif) and ebullitive (Ebu) $CH_4$ emission for the control (C) and warm (W) treatment of our mesocosm experiment. Error bars denote 1 s.e.m. ($n = 4$). Differences between treatments were tested with a $t$-test

our experiment only accounted for 10% (control) and 5% (warm) of total $CH_4$ emissions (Fig. 2), and correlated significantly with temperature during the year (Supplementary Fig. 4). This positive correlation corroborates findings for diffusive $CH_4$ emissions from another mesocosm experiment[33] as well as from an analysis of data from globally distributed wetland and freshwater ecosystems[34]. In part, this may be due to gas dissolution during bubble rise[35]. Interestingly, however, cumulative annual diffusive $CH_4$ fluxes in our experiment show that year-round warming did not enhance the diffusive emission pathway (Fig. 2). This clearly shows that increased $CH_4$ production with warming (as indicated by increased ebullition) does not necessarily result in increased diffusive $CH_4$ emissions over a longer period of time (e.g., a year), likely because of (over)compensation by increased $CH_4$ consumption rates in the sediment top layer and water column[12]. This further substantiates the key role of the ebullitive pathway in total $CH_4$ emissions.

**Increased dominance of ebullition under climate change**. The existence of a threshold temperature marking the onset—or a strong increase—of ebullition is a major concern for sudden shifts

in global GHG emissions with climate warming. We found that above a sediment temperature of ~10 °C (Fig. 3) ebullition strongly increases with temperature, which coincides with findings in another study[35]. However, despite this similarity, the threshold temperature varies among systems (Fig. 1), likely depending on the physical structure of the sediment[25], forcing mechanisms[7], and the methanogenesis rate at a certain temperature. The latter is ultimately determined by the microbial community[36,37] and environmental factors including quality and quantity of available organic substrate[19,24]. With increasing substrate availability, $CH_4$ ebullition shows a stronger temperature dependence than diffusive $CH_4$ emission in northern ponds and lakes[15]. Also, for our experiment, we see that $CH_4$ ebullition becomes the dominant flux above sediment temperatures of ~10 °C (Fig. 3). Our findings thus suggest that climate warming may change diffusion-dominated systems in colder areas into higher-emission, ebullition-dominated ones. This is particularly relevant for boreal and Arctic waters where >50% of the studied systems (with available data on both $CH_4$ ebullition and diffusion) show that diffusion is the most important emission pathway[5]. These cold systems already have an important share in global freshwater GHG emissions as their surface area accounts for almost half of the area of the world's lakes and ponds[4,5]. Many of these systems are small and shallow[38] and will therefore rapidly warm as a consequence of the predicted increase in air temperature, which is expected to be stronger than the global average in the boreal and Arctic region[18,39]. In addition, many of the high northern latitude lakes are formed in permafrost soils. An increase in temperature should therefore not only enhance microbial metabolism of existing substrates, but also lead to more permafrost thaw, which increases substrate availability and methanogenesis per square meter of lake on decadal to century time scales[40], though the process may be limited by widespread thaw lake drainage[41]. The resulting increase in ebullition in these high-latitude systems may greatly amplify their share in freshwater GHG emissions.

## Discussion
In the long term, ebullition rates are primarily driven by the interaction between temperature and substrate supply for methanogenesis[15,20,42]. In systems with low organic matter

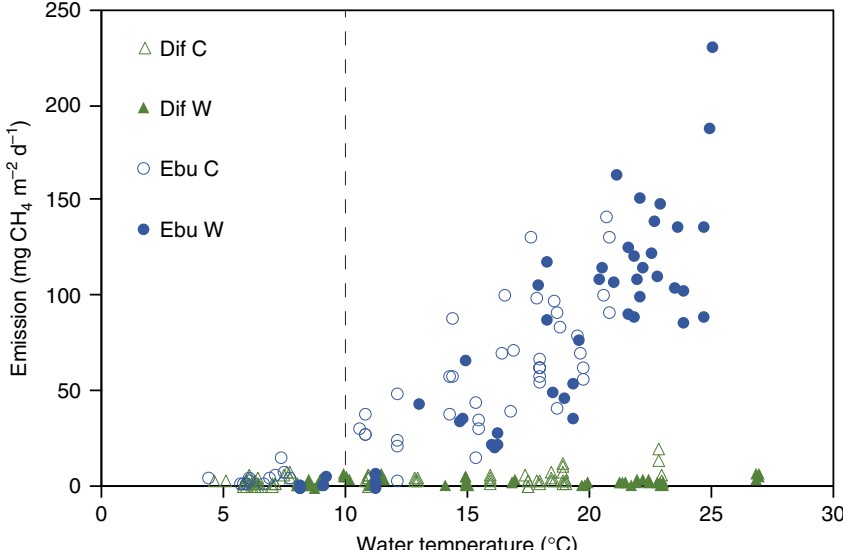

**Fig. 3** Diffusive and ebullitive $CH_4$ emissions expressed against water temperature. Dif refers to diffusive $CH_4$ emissions and Ebu refers to ebullitive $CH_4$ emissions. The dashed vertical line represents the onset of strongly increased ebullition and a transition to ebullition-dominated flux. Open and closed symbols denote control (C) and warm (W) treatment, respectively

production and low allochthonous input, $CH_4$ emission may therefore become substrate-limited as a result of global warming. However, eutrophication and the associated increase in organic matter production may preclude substrate limitation and effectively fuel ebullition[15,27,43–45]. Ongoing eutrophication[46] enhanced by climate change-related increases in sediment nutrient release and organic carbon and nutrient loading from catchments[47–49] will therefore likely boost $CH_4$ ebullition at a global scale. Yet, the strongest increase in ebullition can be expected in shallow waters[15,45,50–52] due to limited stratification, sediment temperature being strongly related to atmospheric temperature[53], and direct solar warming of the sediments[16,54]. Small ponds and shallow lakes, which are the most abundant freshwaters globally[2], are therefore expected to become hot spots of ebullition. They are also among the systems that receive relatively large terrestrial carbon inputs due to their high perimeter-to-volume ratio[17]. The growing number of impoundments[55] also accumulate organic sediments at high rates and are already characterized as ebullition hot spots[20,22].

The high spatiotemporal heterogeneity of ebullition together with paucity of data currently challenge accurate predictions of the absolute increase in emissions as a result of global warming[9]. If the availability of suitable organic matter is not limiting, both the temperature–ebullition relationships fitted on the field data and our experimental manipulation indicate that climate warming will increase $CH_4$ ebullition by 6–20% per 1 °C warming (Fig. 1 and Table 1). It is currently unknown whether relative increases in ebullition will differ among certain climate regions. Microbial communities in the Arctic, for example, are well adapted to low temperatures[56]. However, optimum growth temperatures of cold-adapted methanogens are often much higher than in situ temperatures[56] and $CH_4$ production in permanently cold sediments has been shown to adapt when exposed to higher than in situ temperatures through changes in community composition and metabolic rates[36,57]. Additionally, the temperature dependence of sediment $CH_4$ production does not seem to differ between cold and warm climate regions[10]. Therefore, there is no a priori reason to assume that it would be different for $CH_4$ ebullition. The exponential nature of the empirical relationships in Fig. 1 points out that the absolute effects of warming on ebullition are largest in warmer areas and during the warmest months. Exponential increases in methanogenesis rates have been found up to temperatures of 40 °C[10].

Overall, our experimental results combined with the analysis of field data provide evidence that temperature drives $CH_4$ ebullition in different types of freshwater ecosystems in different parts of the world. This strongly suggests that climate change will substantially increase freshwater $CH_4$ emissions through a disproportional increase in ebullition, forming a positive feedback between climate change and freshwater GHG emissions. Our study is the first to show that a strong relationship between temperature and ebullition exists on a large geographical scale and provides estimates of the increase in $CH_4$ ebullition with climate warming. However, the limited availability of data sets (both globally and per system type) hampers accurate predictions about the relative and absolute increase of this important GHG emission pathway. Hence, to be able to extrapolate our findings to global freshwater $CH_4$ emissions, as well as to better predict changes in future emissions, we stress the need for more measurements including ebullition with high spatiotemporal coverage.

## Methods

**Experimental setup.** Experiments were conducted in eight metal cylindrical 988 L indoor mesocosms called limnotrons with an average depth of 1.35 m and an inner diameter (ID) of 0.97 m (for more information see ref. [58]). The limnotrons were filled with ~70 L of pre-sieved (5 mm mesh size), soft, muddy sediment, and subsequently filled to the top with tap water. Sediment was collected from a mesotrophic shallow pond in Wageningen, The Netherlands (coordinates in DMS: 51°59′16.0″N 5°40′06.1″E) on 13 February 2014. Sediment contained 69 ± 4% (mean ± s.d.) water (% fresh weight) and 6.6 ± 1.1% (mean ± s.d.) organic matter (% dry weight; loss on ignition method). To promote a more diverse initial benthic community, an extra portion of sediment (<8 L) was collected from a nearby eutrophic pond (coordinates in DMS: 51°58′56.7″N 5°43′34.5″E). Water was circulated among limnotrons for 2 days to promote similar starting conditions. Subsequently, nutrients and a phytoplankton inoculum (see below) were added in order to mimic phytoplankton-dominated shallow lakes. The phytoplankton inoculum was obtained by concentrating water from the two ponds mentioned before.

To mimic open water turbulence, we installed two compact axial fans (AC axial compact fan 4850 Z, ebm-papst St. Georgen GmbH & Co. KG, Georgen, Germany) set to an air flow rate of $100 \, m^3 \, h^{-1}$, as well as an aquarium pump (EHEIM compact 300, EHEIM GmbH & Co. KG, Deizisau, Germany) at a depth of 6 cm set at a rate of $150 \, L \, h^{-1}$, which resulted in a piston velocity ($k_{600}$) of 0.44 ± 0.01 (mean ± s.d.) m $d^{-1}$ ($k_{O_2}$ determined in deoxygenated water, after ref. [59]). This $k_{600}$ is typical for small lakes and ponds $(0.001–0.01 \, km^2)$[17]. The limnotrons were carefully filled with demineralized water twice a week to compensate for evaporative losses.

The incident light intensity was constant throughout the experiment with 175 ± 25 (PAR; mean ± s.d.) μmol photons $m^{-2} \, s^{-1}$, provided by two HPS/MH lamps (CDM-TP Elite MW 315–400 W, AGRILIGHT B.V., Monster, The Netherlands). The light:dark cycle followed Dutch seasonality, varying from 8 h of light at midwinter to 17 h at midsummer. Nutrients were added in a way that start concentrations were achieved of 86 ± 19, 2.4 ± 0.8, and 152 ± 37 μM (mean ± s.d.), for nitrate ($NO_3^-$), phosphate ($PO_4^{3-}$), and total silicon (Si), respectively. Nutrient losses by sampling were compensated by weekly additions of nitrate and phosphate. Control mesocosms ($n = 4$) were subjected to a natural seasonal temperature cycle based on the temperature data of Dutch lakes. The warm mesocosms ($n = 4$) followed the same temperature cycle + 4 °C. From 3 to 10 August, a heat wave of an additional +4 °C was simulated in both treatments (Supplementary Fig. 1). Temperature was measured at depths of 0.5 and 1.0 m using PT100 electrodes and logged at 1 min intervals (Specview 32/859, SpecView Ltd., Uckfield, UK). The experiment started on 3 March 2014 and ended on 1 February 2015. Additional information on phytoplankton, zooplankton, and bacterial dynamics in this experiment can be found in refs. [60,61].

**Gross primary production.** Rates of gross primary production (GPP) were determined every week for each mesocosm using the diel oxygen technique[62]. DO and temperature were measured every 15 min for 24 h at a depth of 0.4 m, using a multi-parameter meter (HQ40d, Hach, Loveland, CO, USA) equipped with a luminescent/optical dissolved oxygen (LDO) probe (IntelliCAL LDO101). Probes contain a factory calibration of which its accuracy was checked each week by measurements of 0 and 100% $O_2$-saturated water as well as by comparing readings of the two oxygen probes in a single mesocosm. GPP was calculated following the equations in Table 2 of ref. [62], in which the piston velocity ($k$) was determined in deoxygenated water as mentioned before.

**Sedimented carbon.** The amount of sedimented carbon was calculated by fortnightly (until heat wave) and monthly (after heat wave) analyses, where sedimentation rates were determined by hanging sedimentation traps (9 cm diameter, 18 cm height, and 1.1 L volume) at 1 m depth for a period of 3 days in each limnotron. The contents of the sediment trap were filtered over pre-washed GF/F filters (Whatman, Maidstone, UK), dried at 60 °C overnight and analyzed for carbon on a NC analyser (FLASH 2000 NC elemental analyser, Brechbuehler Incorporated, Interscience B.V., Breda, The Netherlands). To correct for seston particulate organic carbon (POC) in the overlaying water in the sedimentation traps, water samples were taken with a tube-sampler in the middle of the limnotron on the same day as the sedimentation traps were taken out. These seston samples were handled and analyzed in the same manner as the sedimentation samples. Sedimentation rates were calculated by substracting seston POC from the total amount of POC captured in the sediment trap. The cumulative annual sedimented carbon was calculated as the area under the curve of these sedimentation rates.

**Diffusive fluxes.** Diffusive fluxes of $CH_4$ were measured once every 2 weeks (until heat wave) and once every 4 weeks (after heat wave) at the end of the light and the end of the dark period. Additional measurements were performed in the week before, during, and after the heat wave. Diffusive $CH_4$ fluxes were measured over a 3-min period using a cylindrical-shaped transparent acrylic top chamber (ID 29.2 cm; headspace height 18 cm) connected in a closed loop to Greenhouse Gas Analyzers (model GGA-24EP, Los Gatos Research, Santa Clara, CA, USA, and model G2508 CRDS Analyzer, Picarro, Santa Clara, CA, USA). Both devices yielded similar results in a post-experiment comparative test, ensuring consistency of measurements. Fluxes were calculated as described by ref. [63]. Flux measurements were performed in triplicate. During data analysis, replicates influenced by ebullition (causing a sudden increase in $CH_4$ concentration) were removed and the

average flux of the (remaining) replicates was calculated for each limnotron. The diel flux was determined as a weighted average of the light and dark period fluxes, based on day length. Limnotrons were always measured in random order to avoid any time or order effects.

**Ebullitive fluxes**. $CH_4$ release via ebullition (bubble flux from sediment) was estimated by continuously collecting bubbles throughout the experiment, using two bubble traps in each limnotron. Bubble traps consisted of an inverted funnel (ID 15.2 cm) connected to a 120 mL infusion bottle via an 80 cm long tube (ID 10 mm). Funnels were installed ~50 cm below the water surface and tubes and infusion bottles were completely filled with limnotron water. Gas-filled infusion bottles were collected (and immediately replaced) 13 times during the experiment, always before completely being filled with gas. The number of days before collecting depended on the ebullition rate—determined by visual inspection of the gas volume in the bottles—and ranged from 8 to 65 days (median: 21). After collection, the volume of gas was determined by subtracting the weight of each bottle from the pre-determined full-filled weight (i.e., completely filled with water) of the bottle. $CH_4$ concentrations in the gas were measured on an HP 5890 gas chromatograph equipped with a Porapak Q column (80/100 mesh) and a flame ionization detector (Hewlett Packard, Palo Alto, CA, USA). The $CH_4$ content of the collected ebullitive gas ranged from 0 to 95% with a mean of $57 \pm 3\%$ (95% confidence interval). The amount of gaseous $CH_4$ in each bottle was determined by multiplying the $CH_4$ concentration ($C_{gas}$) by the volume of gas ($V_{gas}$). The $CH_4$ in the bottles was assumed to be in equilibrium with the water phase. Hence, the amount of $CH_4$ dissolved in the water ($C_{water} \times V_{water}$) was calculated using Henry's law and its solubility constant for $CH_4$, taking the respective water temperature into account[64]. The total amount of $CH_4$ in each bottle was calculated by summing the aqueous and gaseous content and divided by funnel surface ($A$) and time ($\Delta t$) to calculate $CH_4$ ebullition per square meter:

$$\frac{(C_{gas} \times V_{gas}) + (C_{water} \times V_{water})}{\Delta t \times A} \qquad (1)$$

Ebullitive gas samples were analyzed for $O_2$, $CO_2$, $N_2O$, and $N_2$ once during the experiment, using a gas chromatograph coupled to a mass spectrometer (Agilent 5975C; Agilent, Santa Clara, CA, USA). $O_2$ and $CO_2$ each typically formed less than 1% of total gas volume, while $N_2O$ could not be detected. $N_2$ was the second main component of the gas, together with $CH_4$ composing about 99% of ebullitive gas volume. On several occasions, gas volume in bubble traps was visually inspected before and after diffusive flux measurements to assess whether possible disturbance during this measurement would trigger bubble release. We observed no effect. Between 9 and 15 September, bubble traps were not in use due to maintenance.

**Model approach**. To describe the temperature dependency of $CH_4$ ebullition for each system, we used a modified Arrhenius equation[65]:

$$E_T = E_{20} \times \theta_s^{(T-20)} \qquad (2)$$

Where $E_T$ is the ebullition rate in mg $CH_4$ m$^{-2}$ d$^{-1}$, at temperature $T$ (°C), $E_{20}$ is the ebullition rate in mg $CH_4$ m$^{-2}$ d$^{-1}$ at 20 °C, and $\theta_s$ is the overall system temperature coefficient (dimensionless)[65,66]. The modified Arrhenius expression was fitted on the data using nonlinear regression analysis in IBM SPSS 21 (IBM, Armonk, NY, USA) that uses the Marquardt–Levenberg algorithm, an iterative procedure to find model parameters that minimize the residual sum of squares. Despite its limitations at the lower and upper end of the temperature range, the modified Arrhenius expression is a useful and often applied method of determining temperature dependencies of ecological and microbiological processes[65,66].

**Effect of changing $CH_4$ solubility on $CH_4$ ebullition**. Solubility-adjusted ebullition rates (Supplementary Fig. 3), i.e., the ebullition rates that would have occurred if $CH_4$ solubility in sediment pore water remained constant, were estimated as follows:

$$\frac{V_s \times W_s \times \Delta S_{CH_4}}{\Delta t} + E_{CH_4} \qquad (3)$$

Where, $V_S$ is total sediment volume (95 L m$^{-2}$), $W_S$ is sediment water content (as a fraction; 0.69), $\Delta S_{CH_4}$ is the change in $CH_4$ solubility (mg $CH_4$ L$^{-1}$) at a depth of 1.3 m due to changes in temperature[67] between moment of bubble trap deployment and harvest, $\Delta t$ is amount of days between trap deployment and harvest, and $E_{CH_4}$ is the ebullition rate (mg $CH_4$ m$^{-2}$ d$^{-1}$) in that period. For these calculations, we assumed year-round $CH_4$-saturated pore waters.

**Literature data acquisition**. We performed a standardized literature search on data on Google Scholar and the Data Observation Network for Earth (DataONE) using the keywords $CH_4$, methane, ebullition, bubbling, and bubble flux in all possible combinations in August 2016, similar to ref. [17]. We further expanded our literature collection by adding the relevant references indicated in studies we found.

From all the papers that were collected, we selected those that contained ebullition data from open waters such as ponds, lakes, reservoirs, and rivers within an air/water/sediment temperature range of at least 10 °C to quantitatively model the relationship between ebullition and temperature. Due to the temporal heterogeneity of ebullition, data from short measurements (<24 h) can seriously under- or overestimate ebullitive fluxes and were therefore excluded[7,9]. We excluded saline and brackish waters since their biogeochemistry strongly deviates from freshwater ecosystems. Five papers fulfilled our requisites[15,42,68–70]. We also included unpublished data from a temperate eutrophic city pond, obtained directly from the authors.

For the post-glacial lakes (Sweden)[70], boreal meso-eutrophic forest ponds (Canada)[15], temperate river Saar (Germany)[42], and temperate eutrophic city pond (Netherlands), data were obtained directly from the authors. Raw data from the temperate farm ponds (USA) were obtained from Table 1 of ref. [68]. Data of the subtropical eutrophic city pond (China) was extracted from ref. [69], using WebPlotDigitizer (http://arohatgi.info/WebPlotDigitizer), a web-based tool to obtain high-precision numerical data from plots.

Ebullition data for all studies were obtained by multi-seasonal ebullition measurements, using funnel-type bubble traps. Ebullition was calculated considering the volume of the captured gas and a constant (average) $CH_4$ concentration (refs. [15,42,69]) or the $CH_4$ concentration of each individual gas sample (refs. [68,70], this study, and the unpublished data set). The published data of temperate river Saar was based on an average bubble $CH_4$ content of 80%, which was the average of several measurements[42]. The high-resolution temperature and ebullition data (continuous logging by automated bubble traps[42]) of this system were averaged over 24 h intervals to produce the dots in Fig. 1. For the subtropical eutrophic city pond, a constant $CH_4$ concentration of 43.3% was used, which was based on the concentration in a nearby eutrophic pond[69]. For the boreal meso-eutrophic forest ponds[15], a $CH_4$ concentration of 57.6% was used, which was the average of all gas samples. Measured $CH_4$ concentration ranges for the post-glacial lake data are provided in ref. [39]. The unpublished data of a temperate eutrophic city pond consist of six (three in the littoral (<1 m deep) and three in the central part of the pond (max. 2 m deep)) bubble traps (ID 35 cm), measured over 24 h for each month of the year. Each time for each bubble trap, the collected gas volume was measured and the $CH_4$ concentration was analyzed on a HP 5890 gas chromatograph equipped with a Porapak Q column (80/100 mesh) and a flame ionization detector (GC-FID, Hewlett Packard, Palo Alto, CA, USA). The temperature used in the regression model is the 24-h average water temperature measured with a multi-parameter portable meter (HQ40d, Hach, Loveland, CO, USA) equipped with a luminescent/optical dissolved oxygen (LDO) probe (IntelliCAL LDO101) at the littoral and at the deep station, at ~0.5 m depth. Temperature was recorded every 15 min. Because of the strong variation in methods used in the different studies (e.g., using different measurement frequency and duration, and assuming a fixed $CH_4$ concentration versus accounting for temporal variations in $CH_4$ concentrations of bubbles), we refrained from statistically comparing the different temperature–ebullition models.

Two studies contained separate ebullition data sets of multiple locations[42,69]. The temperature–ebullition relationships of all individual locations are described in Table 1 and the ebullition data from the location with the longest data record was included in Fig. 1. The post-glacial lake data in Fig. 1 is replotted from ref. [70]. It is based on a 6-year data set (2009–2014; total of 10,227 individual flux measurements from multiple locations and depths) from lakes in Stordalen, northern Sweden. Note that it is an extended version of a figure with 4 years of data from the same lakes published in ref. [16]. Because of the vast amount of individual flux measurements, data of post-glacial lakes were binned in 1 °C intervals for plotting in Fig. 1. The modified Arrhenius expression was fitted on the raw data.

**Statistical analysis**. To test for monotonic relationships between water temperature and $CH_4$ diffusion, we used Spearman's rank correlations. Temperature values for $CH_4$ ebullition were obtained by averaging water temperature over each bubble collection period, while for $CH_4$ diffusion; the average daily temperature at day of flux measurement was used. Cumulative annual $CH_4$ ebullition, sedimentation, GPP, and diffusion data were tested for normal distribution using the Shapiro–Wilk's test ($P > 0.05$) and a visual inspection of histograms, normal Q–Q plots and box plots. The Brown–Forsythe test was used to assess homogeneity of variances. Differences between treatments were tested with a Student's $t$ test or—in case assumptions for normality and homogeneity of variance were violated—with a Mann–Whitney $U$ test. All $P$ values mentioned are two-tailed. Statistical analyses were carried out using IBM SPSS 21 (IBM, Armonk, NY, USA).

**Data availability**. The data that support the findings of this study are available from the corresponding author on reasonable request.

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

## Acknowledgements

We thank R. Mendonça and T. van Bergen for their advice and discussions, as well as for providing their unpublished ebullition data set. We also thank F. Rust de Carvalho, F. Xue, and N. Helmsing for their assistance during the mesocosm experiment. The vector map of Fig. 1 was designed by Freepik (www.freepik.com). N.B. was supported by Coordenação de Aperfeiçoamento de Pessoal de Nível Superior (CAPES) process number 3934-13-6 and CAPES (Brazil)/NUFFIC (The Netherlands) project 045/2012. GK's funding was provided by the Leibniz Association - project Landscales. The work of M.V. was funded by the Gieskes-Strijbis Foundation. M.W. and B.F.T. acknowledge funding from Vetenskapsrådet Grant 2007–2547 and Nordic Center of Excellence DEFROST under the Nordic Top-Level Research Initiative. The work of J.W. was financially supported by the German Research Foundation (Grant numbers LO1150/5-2 and LO1150/9-1). T.D. was supported by the Natural Sciences and Engineering Research Council of Canada (NSERC) and the Swiss National Science Foundation. S.K. was supported by Nederlandse Organisatie voor Wetenschappelijk Onderzoek (NWO) Veni Grant 86312012.

## Author contributions

S.K. originally formulated the idea. R.C.H.A. conducted the measurements, collected data from published literature, and analyzed the data. R.C.H.A. and S.K. wrote the manuscript. R.C.H.A., M.V., S.S., G.K. and T.F. mainly took care of the experimental units. M. W., B.F.T., J.W. and T.D. contributed original data. N.B. contributed original, unpublished ebullition data and helped with discussions and advice. L.N.d.S.D. and E.T.H.M.P. provided advice for statistical analyses. N.B., E.v.D., T.F., S.H., G.K., L.P.M.L., E.T.H.M.P., J.G.M.R., L.N.d.S.D., S.S., M.V., D.B.V.d.W., M.W., B.F.T., J.W. and T.D. contributed to revisions of the manuscript.

## Additional information

**Competing interests:** The authors declare no competing financial interests.

