## [Peer Review File · Nature Communications]

Reviewers' Comments:

Reviewer #1:

Remarks to the Author:

This manuscript explores the relationship between temperature and methane bubbling in both the literature and a mesocosm experiment. Overall, I found the concept of the paper and the results to be very intriguing and relatively convincing. The relationship between temperature and bubbling was strong, and consistent among diverse studies. This relationship clearly has implications for the future of the global methane and carbon cycles with respect to wet ecosystems. My main criticism relates to the low number of replicates derived from the literature. The authors lay out their methodology for selecting studies containing both temperature and ebullition data, and the rationale for rejecting studies for their analysis. The end result contains only five published datasets. In comparison, Holgerson and Raymond (2016) *Nature Geoscience* were able to find 47 ebullition datasets (see supplementary material). Were you able to collect as many primary datasets and then prune from that point? My concern is that the study does not actually contain "all" of the published datasets and may be strongly biased. In addition, I wonder whether the authors could attempt to contact additional authors to gain additional access to needed data. This would also alleviate the need to use software to pick the data from graphics. The result of this pruned dataset and subsequent analysis generates a very simple story line, that ebullition increases with temperature. The authors do a nice job of discussing the nuances associated with this story, such as the role of organic matter limitation. However, the title and implications of this paper may be overstated based on the evidence. This work, however, is supported by other literature and builds on very similar work by Yvon-Durocher et al (2017) *Nature Climate Change* (although focusing solely on diffusive emissions). A stronger case and more complete explanation of the highly pruned dataset will be needed prior to publication.

Reviewer #2:

Remarks to the Author:

The manuscript by Aben et al. analyzes data from seven northern hemisphere sites, including one of their own, to show a positive relationship between ebullition and temperature. The authors also conducted a mesocosm experiment to assess the relationship between warming and methane emission modes of diffusion and ebullition, wherein the found ebullition to be a function of temperature, but not diffusion. From these analyses the authors concluded that global warming will disproportionately enhance ebullition, an emission mode that largely bypasses oxidation, thus leading to a positive feedback to climate warming.

The study is intriguing and of broad interest. My main concern is that it is presented in an overly simplistic manner. The authors did not provide an explanation or demonstrate any mechanism by which temperature increase enhances ebullition. It was not clear how much of the ebullition response is due to a decrease in methane solubility under warmer water conditions (direct cause) and how much is due to the indirect cause of temperature increasing net ecosystem productivity, which in turn supplies more substrate for methanogenesis (e.g. Whiting and Chanton 1993 *Nature*). I suspect the later is very important, but would prefer to see the authors present quantitative evidence for this mechanism.

My second concern is that the authors did not address the issue of microbial communities and their adaptations to shifts in temperature. Apparently microbes in temperature ecosystems have very different temperature sensitivities than psychrophilic microbes in arctic and boreal systems (e.g. Zimov et al. 1997 *Science*), so the microbially-mediated response of methanogenesis to an increase in temperature should vary regionally. A more rigorous analysis of literature data on microbial methanogenesis and temperature sensitivity could be provided to help constrain the regional ebullition responses to temperature increases in a future warmer world.

Minor comment: Figure 1 caption. Sentence #5 should probably be moved to the #3 position to improve clarity.

Katey Walter Anthony

Reviewer #3:

Remarks to the Author:

Review of Aben et al. «Cross continental increase in methane ebullition under climate change»

Major conclusion of the paper:

The paper presents original data of methane emissions via diffusion and ebullition from a one-year mesocosm experiment with eight units showing that a 4°C warming led to 57% higher total annual methane ebullition from sediments of a shallow mesotrophic pond. The authors compare these results with a compilation of recent literature data on natural systems covering at least 10°C. Within a temperature range between 0 and about 30°C these natural ebullition fluxes seem to follow a quasi-exponential increase with temperature. Based on the data set from six sites of the Northern Hemisphere and their own experimental data, the authors conclude that a 4°C warming would lead to an increase in the methane ebullition flux from global freshwaters to the atmosphere of 25-273%.

This is an important contribution that aims at better constraining an important methane emission pathway. There is general agreement in the research community that the drivers of methane ebullition should be better quantified and the study provides important new data and a synthesis of the relevant literature. In its present form, however, the paper suffers from inconsistencies in presenting the data and a global upscaling effort that might be misleading, because of severe sampling bias.

Originality

The mesocosm study represents a new and highly valuable approach, which allows for a clear statistical analysis of the annual effect of a 4°C warming on ebullition fluxes. The phenomenon that methane ebullition increases quasi-exponentially with temperature is now well established. The authors acknowledge this with their survey of recent field studies. Del Sontro et al. (2010) *Env. Sci. Technol.* 44: 2419-2425 first documented a quasi-exponential increase of CH₄ ebullition rates in a run-of-river reservoir over a full seasonal cycle (10-17°C).

Quality of evidence

Several factors limit the validity of the conclusion that a 4°C raise in global temperatures would lead to an increase in the CH₄ ebullition from freshwaters to the atmosphere of 25-273%. The paper falls short in explaining the assumptions of this order-of-magnitude estimate for an increase in ebullition rates. Issues that should be addressed in more detail include sampling bias due to limited global coverage, narrow range of systems, the use of exponential data fitting and its limits at the lower and upper end of the observed temperature range.

The global coverage of available CH₄ ebullition data is limited and neglects the specific factors linked to permafrost melting and ice breakup in the arctic and temperature changes affecting tropical and subtropical lakes, ponds and river systems. A serious prediction of the effects of a 4°C temperature increase would have to start with a rather detailed analysis of the global river systems and lakes. See Raymond et al. (2013) *Nature*, 503:355-359 for a global study of aquatic CO₂ emissions.

Published ebullition data span a wide range: A recent review by Deemer et al. (2016) in *Bioscience* 66:949-964 revealed a range of more than three orders of magnitude of observed ebullition fluxes from reservoirs (Figure 1: 1-1000 mg CH₄ m⁻² d⁻¹). The data compiled in the present paper all seem to center around ebullition fluxes of ca. 100-1000 mg CH₄ m⁻² d⁻¹ at 20°C which points to a quite narrow range at the higher end of methane ebullition fluxes (Fig 1, N=7)). By contrast,

Extended Data Figure 3 (caption: N=12) shows 3 systems with high total phosphorus and a cluster of 6 low phosphorus systems and one lonely intermediate system (reader sees N= 10). The authors should therefore provide a table that characterizes the different systems covered in their paper (location, nutrient level, chlorophyll concentrations or productivity, organic carbon concentration in sediments etc.) and they should clean up the discrepancies of the number of systems discussed.

The exponential relations with temperature T (e^{aT} with $0.06 < a < 0.26$) or power laws (T^b with $1.36 < b < 2.98$) between ebullition rates and temperature summarize the available data quite well in the observed temperature range ($T < 30^\circ\text{C}$). Several factors, however, will limit the increase in ebullition beyond a high-temperature break point (limited algal growth, faster aerobic carbon mineralization, substrate limitation etc.). On the other hand, the transition between diffusion-driven emissions to ebullition is a critical lower transition point. Extended Data Figures 1 and 2 to provide a nice basis to determine this important tipping point in the mesocosm experiments.

For the upscaling from this limited set of data to the global scale, two critical questions arise in the context of a 4°C warming at a global scale:

- How large is the fraction of aquatic systems in the cold region dominated so far by diffusive emissions that will turn to ebullition in a warming environment? See also Walter et al. (2006), *Nature*, 443:71-75.
- Is the temperature-sensitivity of tropical and subtropical systems comparable to the temperate and boreal systems represented in the database?

The statistical analysis of the influence of temperature would be more convincing if the authors would more systematically explore the two fitting methods and discuss the biological or physical meaning of the parameters obtained.

Additional comments

Many of the cited references are incomplete (5, 8,15, 16, 23, 27, 29) or contain typos (1, 10, 28).
Figure 1 It remains impossible to evaluate the validity of the relation when no data is shown. All panels should be more clearly linked to their references.
Figure 2 What is IQR? Explain.

Bernhard Wehrli

Reviewers' comments:

Reviewer #1 (Remarks to the Author):

This manuscript explores the relationship between temperature and methane bubbling in both the literature and a mesocosm experiment. Overall, I found the concept of the paper and the results to be very intriguing and relatively convincing. The relationship between temperature and bubbling was strong, and consistent among diverse studies. This relationship clearly has implications for the future of the global methane and carbon cycles with respect to wet ecosystems.

- Thank you for your positive words and also for the other constructive comments that we discuss below.

My main criticism relates to the low number of replicates derived from the literature. The authors lay out their methodology for selecting studies containing both temperature and ebullition data, and the rationale for rejecting studies for their analysis. The end result contains only five published datasets. In comparison, Holgerson and Raymond (2016) Nature Geoscience were able to find 47 ebullition datasets (see supplementary material). Were you able to collect as many primary datasets and then prune from that point?

- We agree that our analysis is based on a limited number of studies. Within the next few years more datasets are very likely to be published, but up to now datasets containing reliable ebullition data in relation to temperature are scarce. Our literature search indeed yielded a large set of over 100 studies containing ebullition data, but in the majority of these cases ebullition was quantified based on short term measurements (minutes to hours) which has been proven to be strongly biased and unreliable, and these studies were therefore discarded from our analysis^{1,2}. Unlike ebullitive fluxes, diffusive fluxes can be estimated relatively well based on short term measurements², likely explaining the higher availability of quantitative estimates of diffusion (Holgerson and Raymond³ found 427 datasets in 25 studies) as compared to ebullition (Holgerson and Raymond³ found 47 datasets in 7 studies which unfortunately failed to meet our requisites). For this revision we repeated our literature research and now also included data repositories using the Data Observation Network for Earth (DataONE). This new search yielded one extra study fulfilling our criteria: a publication by Baker-Blocker et al. 1977⁴. We now included their data to Figure 1. This addition leads to a total number of 11 temperature-ebullition datasets in 5 studies (Table 1 & 2).

My concern is that the study does not actually contain “all” of the published datasets and may be strongly biased.

- See also our previous answer. We agree that the word “all” in the sentence “Using all available multi-seasonal CH₄ ebullition data from literature (...)” may indeed have been misleading as we are referring to only those studies that fulfilled our strict requisites. We therefore removed “all” and more explicitly clarified our method of literature search and criteria for selecting datasets. Also, for reasons of transparency we now provide a complete overview of all 11 datasets from literature that fulfilled our requisites (see table 1), including those that are not presented in figure 1.

In addition, I wonder whether the authors could attempt to contact additional authors to gain additional access to needed data. This would also alleviate the need to use software to pick the data from graphics.

- We fully agree and thank the reviewer for this valuable suggestion; see also our answer to the first comment. We contacted the authors of the papers we previously extracted data from using WebPlotDigitizer. Martin Wik⁵, Jeremy Wilkinson⁶ and Tonya DeSontro⁷ shared their raw data with us which resulted in slightly modified data points in figure 1. We added their contribution to the acknowledgements. The authors of the data from the 'eutrophic hemiboreal lake' are still working on the data themselves, and are therefore unable to share their raw data at this moment⁸. As their original publication is also lacking the individual data points (but only includes the fitted exponential model), the eutrophic hemiboreal lake has now been removed from figure 1. We do, however, still cite them as we recognize the importance of their work. The authors of the 'subtropical eutrophic city pond' paper were also not able to share their data at this point⁹; we therefore kept using the datapoints we retrieved directly from their paper.

The result of this pruned dataset and subsequent analysis generates a very simple story line, that ebullition increases with temperature.

- As a result of your comments and those of the other two reviewers we now include a more in-depth analysis of the processes underlying the temperature-ebullition relationship, providing a mechanistic explanation (lines 92–104). We specifically focused on the importance of changes in methane solubility and the provision of organic carbon to the sediment. Additionally, we have now applied a single model to describe the temperature-ebullition relationship to all datasets (lines 54–58 & table 1). This model—a modified Arrhenius function—is more strongly linked to the underlying ecological processes. We believe that these changes have greatly improved the novelty, information content, and profundity of our story line.

The authors do a nice job of discussing the nuances associated with this story, such as the role of organic matter limitation. However, the title and implications of this paper may be overstated based on the evidence.

- We agree that our conclusions are based on measurements in a limited number of systems. In the revised version we have now removed the estimated 25–173% increase in CH₄ ebullition related to a year-round 4°C warming regime as compared to the current temperate climate temperature regime. Instead we discuss the temperature coefficient (theta, θ) of the modified Arrhenius expression of our study systems. These highlight a 6–20% higher ebullition rate for each 1°C temperature rise (lines 30–32, 146–149 & table 1). We now explicitly mention that these estimates are based on our current knowledge and discuss the limitations of our estimates and the need for more ebullition data with high spatio-temporal coverage (lines 148–152 & 165–170). Although the number of systems that we were able to incorporate in this study is limited, our analysis does still show that a consistent and strong relationship between temperature and methane ebullition occurs on a very large spatial scale (figure 1 & table 1). We therefore feel that the title reflects our findings. We are, however, open to editorial suggestions regarding the title.

This work, however, is supported by other literature and builds on very similar work by Yvon-Durocher et al (2017) Nature Climate Change (although focusing solely on diffusive emissions). A stronger case and more complete explanation of the highly pruned dataset will be needed prior to publication.

- We acknowledge and cite (refs Yvon-Durocher et al. 2014¹⁰ and Yvon-Durocher et al. 2017¹¹) the earlier work focussing on diffusion (lines 107–109). However, as ebullition is often the most important emission pathway from inland waters—frequently ebullition fluxes are an order of magnitude higher than diffusive fluxes (e.g. Bastviken et al. 2011¹²)— and responds in a totally different and disproportionate way to temperature change, we believe our paper conveys an important new message that will be a valuable and essential addition to the literature. Please see our earlier replies with respect to the dataset we used. We would like to stress the fact that we performed a robust literature analysis based on strict and clear *a priori* criteria as in Holgerson & Raymond (2016)³, and were in no way “pruning” data. We have now clarified this further in the manuscript (lines 50–55 & 269–278).

Reviewer #2 (Remarks to the Author):

The manuscript by Aben et al. analyzes data from seven northern hemisphere sites, including one of their own, to show a positive relationship between ebullition and temperature. The authors also conducted a mesocosm experiment to assess the relationship between warming and methane emission modes of diffusion and ebullition, wherein the found ebullition to be a function of temperature, but not diffusion. From these analyses the authors concluded that global warming will disproportionately enhance ebullition, an emission mode that largely bypasses oxidation, thus leading to a positive feedback to climate warming. The study is intriguing and of broad interest.

- Thank you for your positive evaluation and for the nice suggestions addressed below. In the revised version we present additional data on our mesocosm experiment and an additional new in-depth analysis to unravel the mechanisms underlying warming enhanced ebullition (lines 92–104). We also discuss these strongly differing mechanisms now more in detail in the text. Please see our detailed responses below.

My main concern is that it is presented in an overly simplistic manner. The authors did not provide an explanation or demonstrate any mechanism by which temperature increase enhances ebullition. It was not clear how much of the ebullition response is due to a decrease in methane solubility under warmer water conductions (direct cause) and how much is due to the indirect cause of temperature increasing net ecosystem productivity, which in turn supplies more substrate for methanogenesis (e.g. Whiting and Chanton 1993 Nature). I suspect the latter is very important, but would prefer to see the authors present quantitative evidence for this mechanism.

- We indeed agree that it is interesting to elaborate more on the different processes explaining enhanced ebullition with warming. We now include a more detailed analysis of the potential drivers of warming enhanced ebullition in our mesocosms (lines 92–104). We focus on our mesocosm experiment as—in contrast to the literature derived data—we do have all information regarding the sediment characteristics needed to do the calculations. We show that the change in CH₄ solubility with changing temperature had a negligible effect on cumulative annual CH₄ ebullition. However, for the period during which the temperature rose, dissolved CH₄ that turned gaseous contributed 14% (control) and 7% (warm) to the total CH₄ ebullition. For the period with dropping temperatures, CH₄ ebullition was lowered by 17% (control) and 13% (warm) as a result of increased pore water solubility (Supplementary Fig. S3). We do acknowledge that temperature-driven higher ecosystem productivity^{7,13-15} as well as higher sedimentation rates^{16,17} may lead to higher ebullitive rates. However, our warm treatment did not significantly differ from our control with respect to gross primary production and carbon sedimentation (which we now show in Supplementary Figure S2 and included in lines 93–96). These processes are therefore unlikely to have played an important role in the increase in ebullition due to year-round warming. The expected increase in nutrient loading due to climate change^{18,19}, however, and the resulting higher ecosystem productivity are more likely to play an important role in regulating GHG emissions^{7,14,15,20}, as discussed in lines 135–141 of the manuscript.

My second concern is that the authors did not address the issue of microbial communities and their adaptations to shifts in temperature. Apparently microbes in temperature ecosystems have very different temperature sensitivities than psychrophilic microbes in arctic and boreal systems (e.g Zimov et al. 1997 Science), so the microbially-mediated response of methanogenesis to an increase in temperature should vary regionally. A more rigorous analysis of literature data on microbial methanogenesis and temperature sensitivity could be provided to help constrain the regional ebullition responses to temperature increases in a future warmer world.

- We acknowledge the potential differences in temperature dependencies of microbial communities across different climate regions, in the psychrophilic, mesophilic and thermophilic range²¹. We studied literature for information on temperature sensitivities of methanogenesis and we searched for evidence on how this microbial process can be affected by climate warming. Briefly, we found that optimum growth temperatures of psychrophilic Archaea (including methanogens) are often much higher than *in situ* temperatures²². We also found that the methane production capacities in permanently cold sediments adapt to much higher than *in situ* temperatures, possibly due to higher enzymatic activity and a relative increase in mesophilic micro-organisms^{21,23}. The temperature response of sediment methane production in cold climate regions can therefore be expected to be similar to that of warmer climate regions²⁴. We included the discussion on this topic in lines 152–158 of the manuscript.

Minor comment: Figure 1 caption. Sentence #5 should probably be moved to the #3 position to improve clarity.

- Thank you for your suggestion. We now made it more clear that the colored dots denote the experimental data.

Katey Walter Anthony

Reviewer #3 (Remarks to the Author):

Review of Aben et al. «Cross continental increase in methane ebullition under climate change»

Major conclusion of the paper:

The paper presents original data of methane emissions via diffusion and ebullition from a one-year mesocosm experiment with eight units showing that a 4°C warming led to 57% higher total annual methane ebullition from sediments of a shallow mesotrophic pond. The authors compare these results with a compilation of recent literature data on natural systems covering at least 10°C. Within a temperature range between 0 and about 30°C these natural ebullition fluxes seem to follow a quasi-exponential increase with temperature. Based on the data set from six sites of the Northern Hemisphere and their own experimental data, the authors conclude that a 4°C warming would lead to an increase in the methane ebullition flux from global freshwaters to the atmosphere of 25-273%. This is an important contribution that aims at better constraining an important methane emission pathway.

- Thank you for your kind words and valuable comments. We made significant changes in the main text in response to your feedback, which we discuss in detail below. Please note that due to a minor change in which the ebullition rate was calculated (lines 245–251; we now included the CH₄ dissolved in the water in the bottles capturing the bubbles) the 57% increase in total annual methane ebullition changed to 51%.

There is general agreement in the research community that the drivers of methane ebullition should be better quantified and the study provides important new data and a synthesis of the relevant literature. In its present form, however, the paper suffers from inconsistencies in presenting the data and a global upscaling effort that might be misleading, because of severe sampling bias.

We removed inconsistencies in the presentation of our data by omitting temperature-ebullition relationships from our panel where individual data points were not available, and by including measured data we obtained from the authors upon request. Furthermore, for reasons of transparency we now provide an overview of all temperature-ebullition datasets that fulfilled our requisites (table 1 & 2), including those not shown in figure 1. We solved the problem of inconsistent model fits (exponential vs. power law) by applying a single model to describe the temperature-ebullition relationship to all datasets (lines 54–58 & table 1). This model—a modified Arrhenius function—is more strongly linked to underlying ecological processes. We agree that our conclusions are based on measurements in a limited number of systems, and now use the temperature coefficient (θ) of the modified Arrhenius expression to describe the possible increase in methane ebullition with climate warming and to avoid the bias of the (previously used) different model fits. We also discuss in more detail the factors that may have influenced the temperature-ebullition relationships of our literature derived datasets (including sampling method and seasonality related factors; see lines 69–79) and mention the uncertainty in our predicted increase related to the limited datasets (lines 165–170). We now explicitly mention that these estimates are based on our current knowledge and discuss the limitations of our estimates and the need for more ebullition data with high spatio-temporal coverage (lines 148–152 & 165–170). Although the number of systems that we were able to incorporate in this study is limited, our analysis does still show that a consistent and strong relationship between temperature and methane ebullition occurs on a very large spatial scale (figure 1). See also our response to reviewer 2 with respect to the potential differences in temperature dependencies of microbial communities across different climate regions, in the psychrophilic, mesophilic and thermophilic range²¹.

Originality

The mesocosm study represents a new and highly valuable approach, which allows for a clear statistical analysis of the annual effect of a 4°C warming on ebullition fluxes. The phenomenon that methane ebullition increases quasi-exponentially with temperature is now well established. The authors acknowledge this with their survey of recent field studies. Del Sontro et al. (2010) *Env. Sci. Technol.* 44: 2419-2425 first documented a quasi-exponential increase of CH₄ ebullition rates in a run-of-river reservoir over a full seasonal cycle (10-17°C).

- Thank you for your positive words regarding the originality of our study. We have now also included the above-mentioned DelSontro study in our paper when discussing the temperature that onsets a strong increase in ebullition (lines 117–119).

Quality of evidence

Several factors limit the validity of the conclusion that a 4°C raise in global temperatures would lead to an increase in the CH₄ ebullition from freshwaters to the atmosphere of 25-273%. The paper falls short in explaining the assumptions of this order-of-magnitude estimate for an increase in ebullition rates. Issues that should be addressed in more detail include sampling bias due to limited global coverage, narrow range of systems, the use of exponential data fitting and its limits at the lower and upper end of the observed temperature range.

- Instead of the different model fits (power law and exponential), we now use a modified Arrhenius expression to describe the temperature-ebullition relationship for all of the datasets (lines 54–58 & table 1). In the methods section, we discuss the use of the modified Arrhenius expression and its ecological relevance, and mention its limitations at the lower and upper end of the temperature range (lines 309–314). We removed the 25–273% increase with 4°C warming and now use the temperature coefficient (theta, θ) of the fitted Arrhenius expression, which reveals a 6–20% increase in CH₄ ebullition per 1 °C warming. Methodological differences that may have affected the temperature-ebullition relationships are now discussed in lines 69–79. We also specifically mention the uncertainty in our predicted increase of CH₄ ebullition with warming (resulting from the limited datasets and global coverage) in lines 165–170. Lastly, we now more explicitly mention that the observed strong relationship between temperature and ebullition only applies to systems without substrate limitation (lines 66–68, 135–137 & 148–152). It was not our intention to suggest that the observed relationship applies to all systems and hope we made this more clear now in the revised manuscript.

The global coverage of available CH₄ ebullition data is limited and neglects the specific factors linked to permafrost melting and ice breakup in the arctic and temperature changes affecting tropical and subtropical lakes, ponds and river systems. A serious prediction of the effects of a 4°C temperature increase would have to start with a rather detailed analysis of the global river systems and lakes. See Raymond et al. (2013) *Nature*, 503:355-359 for a global study of aquatic CO₂ emissions.

- We appreciate the suggestion on how to make a serious prediction of increased CH₄ ebullition with climate warming. However, because of the limited global coverage of available CH₄ ebullition data as well as issues related to differences in methodology and sampling bias^{1,2}, we have chosen to refrain from attempting to make a generalized quantitative prediction of the global increase in CH₄ ebullition with climate warming. We limit our prediction of relative increase of ebullition with climate warming (6–20% per 1 °C warming) to the findings of our available datasets and discuss its limitations (lines 148–152 & 165–170) as well as factors affecting the temperature-ebullition relationships (lines 69–79) and uncertainties regarding different climate regions (lines 152–158).

Published ebullition data span a wide range: A recent review by Deemer et al. (2016) in *Bioscience* 66:949-964 revealed a range of more than three orders of magnitude of observed ebullition fluxes from reservoirs (Figure 1: 1-1000 mg CH₄ m⁻² d⁻¹). The data compiled in the present paper all seem to center around ebullition fluxes of ca. 100-1000 mg CH₄ m⁻² d⁻¹ at 20°C which points to a quite narrow range at the higher end of methane ebullition fluxes (Fig 1, N=7)). By contrast, Extended Data Figure 3 (caption: N=12) shows 3 systems with high total phosphorus and a cluster of 6 low phosphorus systems and one lonely intermediate system (reader sees N= 10). The authors should therefore provide a table that characterizes the different systems covered in their paper (location, nutrient level, chlorophyll concentrations or productivity, organic carbon concentration in sediments etc.) and they should clean up the discrepancies of the number of systems discussed.

- The observation that the data of our panel are in the high range of ebullition fluxes observed by the review of Deemer et al. (2016)¹⁴ can be attributed to a number of factors. Their dataset of reservoirs contains much deeper systems than our dataset. Ebullition is known to strongly decrease with increasing water depth, as related to the low temperature of deep water limiting methanogenesis, higher methane loss during bubble rise, and high hydrostatic pressure increasing methane solubility in sediment pore waters as well as limiting bubble buoyancy^{7,15,25,26}. Another important explaining factor may be the underestimation of fluxes due to low temporal resolution of measurements^{1,2}. The collected datasets that met our criteria were generally acquired at higher temporal resolution (multiple 24h measurements to continuous logging) than usually applied in field studies. Finally, the locations used in our panel data are most likely relatively rich in organic matter. As we discuss in our manuscript, the effect of temperature on methane ebullition seems to be strongly dependent on substrate availability and becomes obscured in low productive systems⁷ (lines 66-68).
- For reasons of clarity, we now included two tables demonstrating all collected datasets that met our selection criteria (in addition to what is shown in figure 1) as well as information about the temperature-ebullition relationship and additional information characterizing the systems (e.g. nutrient concentration, water depth, location).

The exponential relations with temperature T ($e^{a \cdot T}$ with $0.06 < a < 0.26$) or power laws (T^b with $1.36 < b < 2.98$) between ebullition rates and temperature summarize the available data quite well in the observed temperature range ($T < 30^\circ\text{C}$). Several factors, however, will limit the increase in ebullition beyond a high-temperature break point (limited algal growth, faster aerobic carbon mineralization, substrate limitation etc.). On the other hand, the transition between diffusion-driven emissions to ebullition is a critical lower transition point. Extended Data Figures 1 and 2 to provide a nice basis to determine this important tipping point in the mesocosm experiments.

- Thank you for this useful suggestion. We now evaluated this transition point of diffusion- to ebullition-dominated emissions for our own dataset and compared this with findings from literature (lines 123–133).

For the upscaling from this limited set of data to the global scale, two critical questions arise in the context of a 4°C warming at a global scale:

- How large is the fraction of aquatic systems in the cold region dominated so far by diffusive emissions that will turn to ebullition in a warming environment? See also Walter et al. (2006), *Nature*, 443:71-75.
 - We did not incorporate a quantitative global upscaling, as explained above. However, in light of the previous comment on the transition from diffusion-driven to ebullition-driven fluxes, this is definitely an interesting and important topic to discuss. We therefore used a recent synthesis of Wik et al. (2016)²⁷ to determine the fraction of diffusion vs. ebullition dominated fluxes in lakes and ponds in the cold region (>50° N). Of their 301 datasets, only 29 contained both ebullition and diffusion measurements. Of these 29 datasets, 17 showed higher diffusive than ebullitive CH₄ emissions. Despite the small number of datasets, this shows that many currently, diffusion-dominated systems may turn into higher-emission, ebullition dominated systems. These interesting findings have now been incorporated in the paper (lines 123–133), for which we thank the reviewer.
- Is the temperature-sensitivity of tropical and subtropical systems comparable to the temperate and boreal systems represented in the database?
 - Since our datasets contains no data from true tropical systems (only subtropical), we used the temperature sensitivity of sediment methane production (the source of CH₄ ebullition) as a proxy for the temperature dependence of CH₄ ebullition. A publication of Marotta et al. (2014)²⁴ found no significant difference in the mean temperature sensitivity of sediment CH₄ production between boreal and tropical sediments. Hence, we argue that this likely also applies to CH₄ ebullition (lines 152–158).

The statistical analysis of the influence of temperature would be more convincing if the authors would more systematically explore the two fitting methods and discuss the biological or physical meaning of the parameters obtained.

- We now applied a single (modified Arrhenius) model as this model is more ecologically relevant and allows for direct comparison between systems (lines 54–58 & table 1).

Additional comments

Many of the cited references are incomplete (5, 8,15, 16, 23, 27, 29) or contain typos (1, 10, 28).

- We apologize for these mistakes. We carefully checked and corrected the references.

Figure 1 It remains impossible to evaluate the validity of the relation when no data is shown. All panels should be more clearly linked to their references.

- We now included measured data for all of the panels.

Figure 2 What is IQR? Explain.

- This abbreviation (interquartile range of boxplot) has now been removed, due to the use of bar charts.

Bernhard Wehrli

References

- 1 Maeck, A., Hofmann, H. & Lorke, A. Pumping methane out of aquatic sediments: ebullition forcing mechanisms in an impounded river. *Biogeosciences* **11**, 2925-2938, doi:10.5194/bg-11-2925-2014 (2014).
- 2 Wik, M., Thornton, B. F., Bastviken, D., Uhlbäck, J. & Crill, P. M. Biased sampling of methane release from northern lakes: A problem for extrapolation. *Geophysical Research Letters* **43**, 1256-1262, doi:10.1002/2015GL066501 (2016).
- 3 Holgerson, M. A. & Raymond, P. A. Large contribution to inland water CO₂ and CH₄ emissions from very small ponds. *Nature Geoscience* **9**, 222-226, doi:10.1038/ngeo2654 (2016).
- 4 Baker-Blocker, A., Donahue, T. M. & Mancy, K. H. Methane flux from wetlands areas. *Tellus* **29**, 245-250 (1977).
- 5 Wik, M. Emission of methane from northern lakes and ponds. PhD dissertation. (Stockholm University, Department of Geological Sciences, 2016).
- 6 Wilkinson, J., Maeck, A., Alshboul, Z. & Lorke, A. Continuous seasonal river ebullition measurements linked to sediment methane formation. *Environmental Science & Technology* **49**, 13121-13129, doi:10.1021/acs.est.5b01525 (2015).
- 7 DelSontro, T., Boutet, L., St-Pierre, A., del Giorgio, P. A. & Prairie, Y. T. Methane ebullition and diffusion from northern ponds and lakes regulated by the interaction between temperature and system productivity. *Limnology and Oceanography* **61**, S62-S77, doi:10.1002/lno.10335 (2016).
- 8 Natchimuthu, S. *et al.* Spatio-temporal variability of lake CH₄ fluxes and its influence on annual whole lake emission estimates. *Limnology and Oceanography* **61**, S13-S26, doi:10.1002/lno.10222 (2015).
- 9 Gao, Y. *et al.* Estimation of N₂ and N₂O ebullition from eutrophic water using an improved bubble trap device. *Ecological Engineering* **57**, 403-412, doi:10.1016/j.ecoleng.2013.04.020 (2013).
- 10 Yvon-Durocher, G. *et al.* Methane fluxes show consistent temperature dependence across microbial to ecosystem scales. *Nature* **507**, 488-491, doi:10.1038/nature13164 (2014).
- 11 Yvon-Durocher, G., Hulatt, C. J., Woodward, G. & Trimmer, M. Long-term warming amplifies shifts in the carbon cycle of experimental ponds. *Nature Climate Change* **7**, 209-213 (2017).
- 12 Bastviken, D., Tranvik, L. J., Downing, J. A., Crill, P. M. & Enrich-Prast, A. Freshwater methane emissions offset the continental carbon sink. *Science* **331**, 50, doi:10.1126/science.1196808 (2011).
- 13 Harrison, J. A., Deemer, B. R., Birchfield, M. K. & O'Malley, M. T. Reservoir Water-Level Drawdowns Accelerate and Amplify Methane Emission. *Environmental Science & Technology* (2017).
- 14 Deemer, B. R. *et al.* Greenhouse Gas Emissions from Reservoir Water Surfaces: A New Global Synthesis. *BioScience* **66**, 949-964 (2016).
- 15 West, W. E., Creamer, K. P. & Jones, S. E. Productivity and depth regulate lake contributions to atmospheric methane. *Limnology and Oceanography* **61**, S51-S61, doi:10.1002/lno.10247 (2015).
- 16 Maeck, A. *et al.* Sediment trapping by dams creates methane emission hot spots. *Environmental Science & Technology* **47**, 8130-8137, doi:10.1021/es4003907 (2013).
- 17 Sobek, S., DelSontro, T., Wongfun, N. & Wehrli, B. Extreme organic carbon burial fuels intense methane bubbling in a temperate reservoir. *Geophysical Research Letters* **39**, doi:10.1029/2011GL050144 (2012).
- 18 Jeppesen, E. *et al.* Climate change effects on runoff, catchment phosphorus loading and lake ecological state, and potential adaptations. *Journal of Environmental Quality* **38**, 1930-1941 (2009).

- 19 Moss, B. *et al.* Allied attack: climate change and eutrophication. *Inland Waters* **1**, 101-105, doi:10.5268/IW-1.2.359 (2011).
- 20 Grasset, C., Abril, G., Guillard, L., Delolme, C. & Bornette, G. Carbon emission along a eutrophication gradient in temperate riverine wetlands: effect of primary productivity and plant community composition. *Freshwater Biology* **61**, 1405-1420, doi:10.1111/fwb.12780 (2016).
- 21 Nozhevnikova, A. N., Holliger, C., Ammann, A. & Zehnder, A. Methanogenesis in sediments from deep lakes at different temperatures (2–70 C). *Water Science and Technology* **36**, 57-64 (1997).
- 22 Cavicchioli, R. Cold-adapted archaea. *Nature reviews. Microbiology* **4**, 331 (2006).
- 23 Schulz, S., Matsuyama, H. & Conrad, R. Temperature dependence of methane production from different precursors in a profundal sediment (Lake Constance). *FEMS Microbiology Ecology* **22**, 207-213 (1997).
- 24 Marotta, H. *et al.* Greenhouse gas production in low-latitude lake sediments responds strongly to warming. *Nature Climate Change* **4**, 467-470, doi:10.1038/nclimate2222 (2014).
- 25 Joyce, J. & Jewell, P. W. Physical controls on methane ebullition from reservoirs and lakes. *Environmental & Engineering Geoscience* **9**, 167-178 (2003).
- 26 McGinnis, D. F., Greinert, J., Artemov, Y., Beaubien, S. E. & Wüest, A. Fate of rising methane bubbles in stratified waters: How much methane reaches the atmosphere? *Journal of Geophysical Research* **111**, doi:10.1029/2005jc003183 (2006).
- 27 Wik, M., Varner, R. K., Anthony, K. W., MacIntyre, S. & Bastviken, D. Climate-sensitive northern lakes and ponds are critical components of methane release. *Nature Geoscience*, doi:10.1038/ngeo2578 (2016).

Reviewers' Comments:

Reviewer #1:

Remarks to the Author:

I find that the authors have done a great job of addressing the comments, and that the narrative of the paper is now more complete. The only substantive editorial comment I have is that both the "Results" and "Discussion" sections are blended into mostly discussion material. Results should be clearly distinguished from the interpretation.

Reviewer #2:

Remarks to the Author:

I found the manuscript to be substantially improved and to now be a compelling and well-written paper. The authors have adequately addressed my concerns, and seem to have fairly addressed the concerns of other reviewers as well.

After reading the revised manuscript, I did however, find a few minor issues could be clarified to further improve the paper.

L125. Please clarify if 10deg C threshold refers to sediment or air temperature, and what time scales does temperature need to be above 10 deg C in order for ebullition to dominate emissions? In lines 101-104, the authors suggested the temperature response is that of microbial metabolism and growth. Is there any kind of hysteresis observed in emissions between rising vs. falling temperatures across the 10 deg C threshold, and how can this be explained by microbial growth patterns?

L. 130-133, here, in the discussion of boreal and arctic lakes increasing in temperature, it could also be mentioned that many of the high northern latitude lakes are formed in permafrost soils, so an increase in temperature should not only enhance microbial metabolism of existing substrates, but it should also lead to more permafrost thaw, which increases substrate availability and methanogenesis per square meter of lake on decadal to century time scales after thaw (see Kessler et al. 2012 JGR Biogeosciences).

L.18. Affiliation #8 is missing the country (Germany).

Reviewer #3:

Remarks to the Author:

The different comments and suggestions of the reviewers have been carefully addressed. Specific improvements include

- 1) better documentation of the data,
- 2) consistent and meaningful model to evaluate the temperature dependency of ebullition,
- 3) careful discussion of the significance and limits of the results obtained.

The paper is a significant contribution to a highly relevant biogeochemical process and it now meets high scholarly standards.

REVIEWERS' COMMENTS:

Reviewer #1 (Remarks to the Author):

I find that the authors have done a great job of addressing the comments, and that the narrative of the paper is now more complete. The only substantive editorial comment I have is that both the "Results" and "Discussion" sections are blended into mostly discussion material. Results should be clearly distinguished from the interpretation.

- We thank the reviewer for the appreciation of our work. We agree the "Results" and "Discussion" sections are rather blended which resulted from following the guidelines of Nature Geoscience (Letter format). We used the Nature manuscript transfer facility – requiring no reformatting - to transfer the manuscript to Nature Communications. We hope we can keep the current format, but if the editors consider restructuring is necessary we will certainly do so.

Reviewer #2 (Remarks to the Author):

I found the manuscript to be substantially improved and to now be a compelling and well-written paper. The authors have adequately addressed my concerns, and seem to have fairly addressed the concerns of other reviewers as well.

- Thank you for your kind words and positive assessment.

After reading the revised manuscript, I did however, find a few minor issues could be clarified to further improve the paper.

L125. Please clarify if 10deg C threshold refers to sediment or air temperature, and what time scales does temperature need to be above 10 deg C in order for ebullition to dominate emissions? In lines 101-104, the authors suggested the temperature response is that of microbial metabolism and growth. Is there any kind of hysteresis observed in emissions between rising vs. falling temperatures across the 10 deg C threshold, and how can this be explained by microbial growth patterns?

- We clarified that 10 °C refers to sediment temperature (lines 132-133).

- The temporal resolution of our measurements did not allow us to exactly determine how long temperature needs to exceed 10 °C in order for ebullition to dominate emissions in our experiment. In general it probably very much depends on system characteristics (e.g. sediment structure, methanogenesis rates, forcing mechanisms) at which exact temperature ebullition onsets and how long this process takes. Additionally, the system's history may be important, as the temperature and duration of the < 10 °C period likely influences the size of

the sediment's existing gas stock, in turn affecting the onset of ebullition. Besides these factors influencing onset of ebullition, domination of total emissions by ebullition is also dependent on the size of the diffusive flux. More research is needed to reveal how omnipresent this 10 °C threshold is.

- We specifically looked for hysteresis comparing thetas between rising and falling temperatures, but could not find evidence for hysteresis. The confidence intervals of the theta's overlapped for both periods, this is also the case when we correct for changes in solubility. We argue that the absence of evidence for hysteresis in our data may be due to the interplay of changes in organic matter availability & quality, methane solubility as well as in microbial community composition, biomass & microbial process rates.

L. 130-133, here, in the discussion of boreal and arctic lakes increasing in temperature, it could also be mentioned that many of the high northern latitude lakes are formed in permafrost soils, so an increase in temperature should not only enhance microbial metabolism of existing substrates, but it should also lead to more permafrost thaw, which increases substrate availability and methanogenesis per square meter of lake on decadal to century time scales after thaw (see Kessler et al. 2012 JGR Biogeosciences).

- Thank you for this useful suggestion. We now included this in lines 147-151.

L.18. Affiliation #8 is missing the country (Germany).

- We now added the country (Germany) to affiliation #8.

Reviewer #3 (Remarks to the Author):

The different comments and suggestions of the reviewers have been carefully addressed. Specific improvements include

- 1) better documentation of the data,
- 2) consistent and meaningful model to evaluate the temperature dependency of ebullition,
- 3) careful discussion of the significance and limits of the results obtained.

The paper is a significant contribution to a highly relevant biogeochemical process and it now meets high scholarly standards.

- Thank you for your positive words.

We would like to thank all reviewers for their constructive feedback, which enabled us to significantly improve our story.